# Prevention and Therapy of Metastatic HER-2^+^ Mammary Carcinoma with a Human Candidate HER-2 Virus-like Particle Vaccine

**DOI:** 10.3390/biomedicines10102654

**Published:** 2022-10-20

**Authors:** Francesca Ruzzi, Arianna Palladini, Stine Clemmensen, Anette Strøbæk, Nicolaas Buijs, Tanja Domeyer, Jerzy Dorosz, Vladislav Soroka, Dagmara Grzadziela, Christina Jo Rasmussen, Ida Busch Nielsen, Max Soegaard, Maria Sofia Semprini, Laura Scalambra, Stefania Angelicola, Lorena Landuzzi, Pier-Luigi Lollini, Mette Thorn

**Affiliations:** 1Alma Mater Institute on Healthy Planet and Department of Experimental, Diagnostic and Specialty Medicine (DIMES), University of Bologna, 40126 Bologna, Italy; 2Department of Molecular Medicine, University of Pavia, 27100 Pavia, Italy; 3ExpreS2ion Biotechnologies, SCION-DTU Science Park, 2970 Hørsholm, Denmark; 4Experimental Oncology Laboratory, IRCCS Istituto Ortopedico Rizzoli, 40136 Bologna, Italy

**Keywords:** breast cancer, vaccine, virus-like particles (cVLP), HER-2, tyrosine kinase receptor, target therapies, cancer immunotherapy, metastasis

## Abstract

Vaccines are a promising therapeutic alternative to monoclonal antibodies against HER-2^+^ breast cancer. We present the preclinical activity of an ES2B-C001, a VLP-based vaccine being developed for human breast cancer therapy. FVB mice challenged with HER-2^+^ mammary carcinoma cells QD developed progressive tumors, whereas all mice vaccinated with ES2B-C001+Montanide ISA 51, and 70% of mice vaccinated without adjuvant, remained tumor-free. ES2B-C001 completely inhibited lung metastases in mice challenged intravenously. HER-2 transgenic Delta16 mice developed mammary carcinomas by 4–8 months of age; two administrations of ES2B-C001+Montanide prevented tumor onset for >1 year. Young Delta16 mice challenged intravenously with QD cells developed a mean of 68 lung nodules in 13 weeks, whereas all mice vaccinated with ES2B-C001+Montanide, and 73% of mice vaccinated without adjuvant, remained metastasis-free. ES2B-C001 in adjuvant elicited strong anti-HER-2 antibody responses comprising all Ig isotypes; titers ranging from 1–10 mg/mL persisted for many months. Antibodies inhibited the 3D growth of human HER-2^+^ trastuzumab-sensitive and -resistant breast cancer cells. Vaccination did not induce cytokine storms; however, it increased the ELISpot frequency of IFN-γ secreting HER-2-specific splenocytes. ES2B-C001 is a promising candidate vaccine for the therapy of tumors expressing HER-2. Preclinical results warrant further development towards human clinical studies.

## 1. Introduction

The human epidermal growth factor receptor-2 (HER-2) is a surface receptor-like tyrosine kinase which plays an oncogenic role in human breast carcinoma and in a fraction of many other tumor types, including bladder, colorectal, lung, stomach and musculoskeletal cancers [1,2]. HER-2 expression is also found in mammary carcinomas and other tumor histotypes of cats [3,4,5] and dogs [6], thus indicating that it is a common hallmark of mammalian carcinogenesis. In human breast cancer, amplification of the HER-2 gene and/or overexpression of its protein product, p185, occurs in 20–25% of cases, and is associated with a poor prognosis [2,7,8].

Therapeutic targeting of HER-2 with monoclonal antibodies, like trastuzumab and pertuzumab, has revolutionized the therapy of breast cancer, leading to significant improvements in tumor response rates and patients’ survival [9,10,11].

Inhibition of surface tyrosine kinases with monoclonal antibodies offers distinct advantages over small molecule tyrosine kinase inhibitors (TKI), because immunoglobulin (Ig) molecules, which are almost as big as p185, can inhibit the dimerization of HER-2, which is a prerequisite to the cross-phosphorylation of intracellular domains, thus blocking mitogenic signal transduction [12,13]. Also, receptor internalization and/or degradation induced by anti-HER-2 mAb binding interferes with HER-2 signaling [14].

Furthermore, antibody binding activates cytotoxic mechanisms mediated by the Fc domain of the Ig molecule, which include complement-mediated cytotoxicity (usually not overly efficient against solid tumor cells), complement-dependent, cell-mediated cytotoxicity by macrophages expressing C3 complement receptors, and antibody-dependent cell-mediated cytotoxicity by natural killer (NK) cells [15,16].

However, not all monoclonal antibodies exert equally well all anti-tumor activities, which depend on the specific HER-2 antigenic epitope bound (for example, pertuzumab is much better at inhibiting dimerization than trastuzumab), and on the isotype of the Ig molecule (for example, mouse IgG2a, IgG2b and IgG3 mediate complement-dependent cytotoxicity better than IgG1) [17,18,19,20]. Furthermore, as also happens with single-drug therapeutic regimes, tumor cells can develop resistance to specific inhibitory mechanisms playing a key role in the therapeutic activity of a single monoclonal antibody [21,22].

To obtain a wider gamut of tumor-inhibitory activities, one strategy is to combine two monoclonal antibodies, as was done with trastuzumab and pertuzumab [12,23]. A more physiological alternative would be to elicit polyclonal antibody responses in patients using vaccines encoding the extracellular domain (ECD) of human HER-2 [24,25,26]. To this end, powerful vaccines are required to obtain broad antibody specificities and multiple Ig isotypes, covering all known anti-tumor activities.

The human HER-2 vaccine candidate developed at Expression Biotechnologies is based on a proprietary virus-like particle (VLP) platform which allows the assembly of ~50 molecules of HER-2 extracellular domain (ECD) on the surface of each particle [27]. A prototypic HER-2-VLP vaccine showed promising therapeutic activity in human HER-2 transgenic mouse models [24]. Now the vaccine has been re-engineered for human administration, scheduled for the first half of 2024; we present here the results of preclinical mammary carcinoma prevention and therapy in mice.

## 2. Materials and Methods

### 2.1. Mice

FVB female mice (6–8 weeks old) were purchased from Charles River (Calco, Lecco, Italy).

Delta16 mice (FVB background) harbor a heterozygous Δ16 transgene, an activated isoform of HER-2 derived from the skipping of exon 16 which caused an in-frame loss of 16 amino acids in the extracellular domain (ECD) [28]. HER-2Δ16 is expressed by human breast cancer and is immunologically cross-reactive with full-length HER-2 [29]. Delta16 mice were bred in our animal facilities by crossing FVB female mice and heterozygotic Delta16 male mice. Transgene-bearing mice were selected by polymerase chain reaction (PCR). Genomic DNA was isolated by digestion of ear tissue (1–2 mm) and extracted as previously described [30]. To detect Δ16 transgene, two sets of primers were used: forward GGT CTG GAC GTC CCA GTC TGA and reverse GAT AGA ATG GCG CCG GGC CTT (Invitrogen, Milan, Italy). Gene expression was evaluated by PCR using Platinum TAQ Polymerase Reactions Kit (Thermo Fisher Scientific, Monza, Italy) and VeritiPro thermal cycler (Thermo Fisher Scientific). Electrophoresis on a 2% agarose gel was performed to detect gene bands in the amplified DNA.

All mice were monitored daily and weighed twice weekly. All in vivo experiments were performed according to Italian and European laws and were authorized by the Italian Ministry of Health (letter 714-2017).

### 2.2. Design, Expression and Purification of Catcher-HER-2 Antigen

The HER-2 ECD comprising amino acids 23–652 (Gene ID: NP_004439) was designed with split protein Catcher sequence at the N-terminus and a GGS linker inserted between HER-2 ECD and Catcher. The antigen construct had an N-terminally BIP secretion signal and a 4 amino acid purification tag, C-tag, at the C-terminus.

The final Catcher-HER-2-C-tag gene sequence was codon optimized for expression in *Drosophila melanogaster* S2 insect cells and subcloned into the proprietary expression plasmid pExpreS2-1 (ExpreS2ion Biotechnologies) and synthesized by GeneArt (Thermo Fisher Scientific).

The ExpreS2 platform (ExpreS2ion Biotechnologies) was used to produce all proteins. Briefly, ExpreS2 Cells were transfected using ExpreS2 Insect-TRx5 transfection reagent (ExpreS2ion Biotechnologies). Thereafter, a polyclonal cell line was selected over a twenty-day period in EX-CELL 420 insect cell medium (Sigma) supplemented with 10% fetal bovine serum, FBS (Thermo Fisher Scientific), and zeocin (Thermo Fisher Scientific). Following selection, the polyclonal cell line was expanded for production in EX-CELL 420 medium. Harvesting of the cell culture supernatant, containing the secreted protein of interest, was done by centrifugation (Beckman Avanti JXN-26, Fixed Angle rotor J-LITE JLA-8.1000) and filtration (0.2 µm vacuum filters, PES) three days after the final spilt. Recombinant Catcher-HER-2-C-tag was purified on a Capture Select C-tag (Thermo Fisher Scientific) affinity column and polished on a preparative Superdex 200pg 26/600 SEC column (Cytiva) equilibrated in 1x PBS (Phosphate buffered saline, Gibco) and eluted in the same buffer.

### 2.3. Design, Expression and Purification of Tag-VLP

The proprietary peptide-binding Tag was added to the N-terminus of the Acinetobacter phase AP205 coat protein (Gene ID: 956335) with a linker sequence (GSGTAGGGSGS) in between. The gene sequence was inserted into the pET28a(+) vector (Novagen) and expressed in BL21 (DE3) competent E. coli cells (New England Biolabas) according to manufacturer’s protocols, and purified as described previously [29].

### 2.4. Formulation and Purification of ES2B-C001 Vaccine

The Tag-VLP and the Catcher-HER-2 were mixed in a 2:1 molar ratio (AP205 subunit per antigen) in PBS containing 10 mM Tris and 200 mM sucrose for 20 h at room temperature, resulting in covalent linkage of the HER-2 antigen to the VLP via the Catcher-Tag system. Conjugated HER-2-VLP was purified by tangential flow filtration using 300 kDa molecular weight cut-off (MWCO) T-series Centramate TFF cassette (Pall). Unbound HER-2 antigen was removed by diafiltration against PBS containing 10 mM Tris and 200 mM sucrose and ultrafiltration served to concentrate the HER-2-VLP to final concentration. To determine coupling efficiency, calculated as percentage conjugation (number of bound antigens divided by total available binding sites (=180) per VLP), densitometric analysis of SDS-PAGE gels were performed.

### 2.5. Vaccinations

ES2B-C001 cVLP vaccine was formulated with Montanide ISA 51 (referred as Montanide) (Seppic, Courbevoie, France) or with PBS (Thermo Fisher Scientific) in a 50/50 volume ratio; control groups received PBS alone. Montanide was emulsified with ES2B-C001 according to the manufacturer’s protocol. The standard dose of vaccine was 10 μg per mouse per administration; in one experiment graded doses ranging 5–40 μg were used. Vaccinations were administered intramuscularly (i.m.) in the hind left leg every two weeks. The total number of vaccinations is detailed in the Results section for each experiment.

### 2.6. Cells

D16-BO-QD cell line (QD for short) was established in our laboratory from a transgenic HER-2-positive mammary carcinoma of a Delta16 female mouse. QD cell line was cultured in MammoCult medium (STEMCELL Technologies, Vancouver, Canada) supplemented with 1% fetal bovine serum (FBS), 100 U/mL penicillin and 10 μg/mL streptomycin (all from Thermo Fisher Scientific).

Human HER-2^+^ breast cancer cell line BT-474 (HER-2^+++^ cell line) and its trastuzumab-resistant clone C5 [24] were routinely cultured in RPMI (Thermo Fisher Scientific) supplemented with 10% FBS, 100 U/mL penicillin and 10 μg/mL streptomycin.

Cells were cultured at 37 °C in a humidified 5% CO_2_ atmosphere and were split once or twice a week according to density using 0.05% trypsin EDTA (Thermo Fisher Scientific).

HER-2 expression of all cell lines is shown in Appendix A.

### 2.7. Local Tumor and Metastasis Therapy Models

In the local tumor therapy model, 7-weeks-old virgin FVB female mice were challenged with 10^6^ QD cells in the mammary fat pad (i.m.f.p.) of the fourth mammary area. Vaccine with or without adjuvant was administered starting two weeks after cell injection. Tumor dimensions were measured with calipers twice a week; tumor volume was calculated as (π/6)(√*ab*)^3^ where *a* = maximal tumor diameter and *b* = maximal tumor diameter perpendicular to *a*. Mice were euthanized if they showed any sign of distress or if tumor volume exceeded 2.5 cm^3^.

In the metastasis therapy model, 9-weeks-old FVB female mice were challenged with an intravenous (i.v.) injection of 10^6^ QD cells in a lateral tail vein, while Delta16 female mice (6–8 weeks old) received 0.25 × 10^6^ QD cells i.v. Mice were euthanized 11–13 weeks after cell injection, an accurate necropsy was performed, lungs were perfused with black India ink and fixed in a modified Fekete’s solution. Metastases were counted under a dissection microscope.

Serum samples from vaccinated and control mice were collected periodically and stored at −80 °C.

### 2.8. Indirect Immunofluorescence and Flow Cytometry

To evaluate the HER-2 cell lines expression level, all the cell lines were incubated with mouse anti-human HER-2 monoclonal antibody MGR2 (Enzo Life Sciences, Farmingdale, NY, USA), then with an anti-mouse IgG labeled with Alexa Fluor 488 (Thermo Fisher Scientific). For the relative quantitative evaluation of anti-HER-2 antibodies elicited by vaccination in the differently treated mice groups, mouse sera were diluted 1:65 and applied to stain a standard human HER-2^+^ cell line (BT-474) followed by addition of anti-mouse IgG-AF488; a sample incubated with the anti-HER-2 monoclonal antibody MGR2 was included in each session to normalize variations in HER-2 expression of BT-474 cells.

### 2.9. Enzyme-Linked Immunosorbent Assay (ELISA)

Serum anti-HER-2 IgG were quantitatively measured in a specific ELISA. Human HER-2 ECD comprising amino acids 22–652 (Gene ID: NP_004439), codon optimized for *Drosophila melanogaster* S2 insect cells and produced in ExpreS2 (ExpreS2ion Biotechnologies) as described above, with a C-terminal Strep-tag II for affinity purification, was used as coating antigen. It was prepared following manufacturer instructions.

Immunoplate Nunc Maxisorp 96-well microplates (Merck, Darmstadt, Germany) were coated overnight with the HER-2 ECD at 1 μg/mL in Carbonate Bicarbonate buffer (Merck). A standard curve (0.04 to 30 ng/mL) with mouse monoclonal antibody H2M5B against human HER-2 (IgG1, R&D Systems, Minneapolis, MN, USA) was run in parallel. The following horseradish peroxidase (HRP)-labeled goat anti-mouse Ig antibodies, all from Thermo Fisher Scientific, were used for detection: total Ig (1:30,000 dilution), IgM (1:10,000 dilution), IgG1 (1:10,000 dilution), IgG2a (1:5000 dilution), IgG2b (1:10,000 dilution), IgG3 (1:5000 dilution).

### 2.10. Enzyme-Linked Immunospot (ELISpot)

FVB female mice (*n* = 4) received two administrations of ES2B-C001+Montanide. The vaccine emulsion was administered bi-weekly at the dose of 10 μg/mouse, and spleens were resected 13 days after the last immunization. Positive control mice (*n* = 2) received three daily intraperitoneal (i.p.) administrations of recombinant mouse interleukin (IL)-12 (provided by S. Wolf, Genetics Institute, Andover, MA). IL-12 was administered as previously reported [31]. Spleens were collected one hour after last IL-12 injection. Control mice (*n* = 2) were untreated.

Red blood cells were removed using Red Blood Cell Lysis Solution (Miltenyi Biotec, Bergisch Gladbach, Germany). ELISpot Mouse IFN-γ kit (R&D systems) was used to perform the analysis. A total of 0.2 × 10^6^ splenocytes/well were seeded and stimulated with HER-2 peptide pool (PepMix, JPT Peptide Technologies GmbH, Berlin, Germany) or left untreated for 48 h. HER-2 peptide pool was solved in dimethyl sulfoxide (DMSO, Merck) and an in vitro splenocytes restimulation with DMSO was performed as control (referred to as vehicle) ELISpot assay was performed according to the manufacturer’s protocol. Spots were counted under a dissection microscope.

### 2.11. Agar Colony Growth Inhibition

In vitro sensitivity of BT-474 (trastuzumab-sensitive) and BT-474 C5 (trastuzumab-resistant) cells to HER-2-VLP-induced antibodies was evaluated in three-dimensional cultures. Cells were seeded at 500 cells/well in 24-well plates in RPMI + 10% FBS + 0.33% agar (Lonza Bioscience Solutions, Siena, Italy) with mouse sera diluted 1:100 or a pharmacologically relevant concentration of trastuzumab (50 μg/mL, kindly provided by Genentech). Colonies (diameter > 90 μm) were counted under an inverted microscope in dark-field 18–30 days after seeding.

## 3. Results

### 3.1. Therapy of Mammary Carcinoma in FVB Mice

The therapeutic activity of ES2B-C001 was tested against mammary carcinomas induced by the injection of QD cells in the mammary fat pad of syngeneic FVB female mice. Vaccinations with ES2B-C001, alone or formulated in the Montanide ISA-51 adjuvant, started 2 weeks after cell challenge and were administered 7 times at bi-weekly intervals, mimicking the expected schedule in patients.

All mice vaccinated with ES2B-C001+ISA 51 were tumor-free at 10 months post-challenge (Figure 1), whereas all mice treated with vehicle alone developed progressive tumors within 1–2 months. Vaccination without adjuvant blocked tumor growth in 70% of mice.

### 3.2. Therapy of Lung Metastases in FVB Mice

A major clinical application of anti-HER-2 vaccines would be against breast cancer metastases, for example to prevent metastasis outgrowth in an adjuvant setting. To model this situation, we injected QD cells intravenously (i.v.) in FVB mice, and one week later we started vaccinations with ES2B-C001.

Eleven weeks after challenge, all mice that were vaccinated with ES2B-C001, with or without adjuvant, were completely devoid of lung metastases, whereas all non-vaccinated mice had hundreds of metastatic lung nodules (Figure 2).

### 3.3. Anti-HER-2 Antibody Response

Serum samples were collected from all treated mice before QD cell challenge and before each vaccination (i.e., two weeks after the previous vaccination). Anti-HER-2 immunoglobulins were analyzed for binding to native HER-2 molecules on the surface of human breast cancer cells BT-474 by means of indirect immunofluorescence followed by flow cytometry or by binding to HER-2 extracellular domain (ECD) in ELISA. Both assays showed that vaccinations elicited strong anti-HER-2 antibody responses, in particular when administered with Montanide (Figure 3).

Serum anti-HER-2 Ig of FVB mice plateaued in the 1–10 mg/mL range, and antibody levels persisted in this range for many months after the last vaccination (Figure 3), indicating that vaccinations elicited long-term immunological memory against HER-2.

ES2B-C001 elicited all subclasses of IgG, including those endowed with the strongest anti-tumor activity, i.e., IgG2a, IgG2b and IgG3 (Figure 4).

### 3.4. Inhibition of Human Breast Cancer Cells in 3D Culture

To better appraise the therapeutic activity of antibodies elicited by ES2B-C001, we studied their inhibitory activity on human HER-2^+^ breast cancer cells BT-474 (trastuzumab-sensitive) and on the trastuzumab-resistant clone BT-474-C5, growing as three-dimensional (3D) colonies in agar, which represent a better model of tumor architecture than 2D cultures. As expected, trastuzumab, applied in a similar concentration as the immune sera, inhibited the growth of BT-474, but not of BT-474-C5. In striking contrast, the antibodies elicited by ES2B-C001+Montanide strongly inhibited both trastuzumab-sensitive and trastuzumab-resistant breast cancer cells (Figure 5), thus providing strong evidence of the potential advantages of a polyclonal antibody response.

### 3.5. Cytokines and T Cell Responses

T cell responses were much less evident than antibody responses; serum cytokine profiles, including gamma interferon (IFN-γ), tumor necrosis factor alpha (TNF-α), and interleukin (IL-) 1β, 2, 4, 5, 6, 10 and 12(p70), as evaluated by BioPlex two weeks after one or two vaccinations, were mostly indistinguishable from those of non-vaccinated mice (data not shown), thus indicating that the vaccine did not elicit a systemic cytokine storm. No relevant peaks of IFN-γ were detected by ELISA from a few hours to two weeks after vaccination.

ELISpot assay revealed the presence of some IFN-γ secreting splenocytes (20.7 ± 2.9 spots/2 × 10^5^ cells) after two in vivo vaccinations followed by in vitro restimulation with a pool of HER-2 ECD peptides (Figure 6).

### 3.6. Prevention of Mammary Carcinoma Onset in Transgenic Mice

Delta16 transgenic mice express an activated isoform of human HER-2 under the transcriptional control of a mouse mammary tumor virus long terminal repeat, leading to the development of multiple mammary carcinomas in all female mice during the first year of life. Delta16 mice are immunologically tolerant to human HER-2, hence the induction of anti-HER-2 immunity in these mice entails a break of tolerance similar to what must occur in human patients to develop protective immune responses against HER-2^+^ tumors.

Young, tumor-free female Delta16 mice were vaccinated twice with varying doses of ES2B-C001 and Montanide, which elicited a strong and persistent anti-HER-2 antibody response (Figure 7A). Of note, peak anti-HER-2 antibody levels of Delta16 mice were lower than those of FVB mice (compare Figure 3 and Figure 7), in keeping with the immunological tolerance to human HER-2 of the former. Antibody titers remained at very high levels for several months after the last immunization, showing that ES2B-C001 induced long-lasting anti-HER-2 immunity also in these mice (Figure 7A).

Vaccination prevented mammary carcinoma onset in 95% of mice, which remained tumor-free until at least 1 year of age. In contrast, all untreated mice succumbed to multiple mammary carcinomas between 4 and 8 months of age (Figure 7B). The results indicate that immunity elicited by ES2B-C001 could effectively block mammary carcinogenesis driven by human HER-2.

### 3.7. Therapy of Lung Metastases in Delta16 Mice Activity

To determine whether ES2B-C001 could be therapeutically effective in Delta16 mice, young females were challenged intravenously with QD cells and then vaccinated every two weeks, starting one week after cell challenge. Metastasis outgrowth in Delta16 mice was blocked by vaccination. All control mice challenged with QD cells had lung nodules (mean ± SEM 68 ± 20 at 13 weeks after challenge), whereas all mice vaccinated with E2SB-C001+Montanide were metastasis-free; 73% of mice vaccinated without adjuvant were also metastasis-free, the remaining had just 1–2 lung nodules (Figure 7C).

### 3.8. Safety Profile of ES2B-C001

The main aim of the present study was to investigate the therapeutic activity of ES2B-C001 against mammary carcinomas, but we also had the opportunity to preliminarily evaluate its safety in mice, in view of formal toxicological studies to be conducted separately in other animal species. Weight curves of unvaccinated and vaccinated mice overlapped (Appendix A). As judged from the constant monitoring of the mice, ES2B-C001 either alone or formulated in Montanide ISA 51 does not compromise the general health status of mice.

## 4. Discussion

ES2B-C001, a VLP-based vaccine being developed for human breast cancer therapy, displayed strong immunogenicity and powerful anti-tumor and anti-metastatic activities in preclinical mouse models of HER-2^+^ mammary carcinoma. ES2B-C001 elicited specific antibody responses comprising all Ig isotypes, which persisted for many months after the last vaccination, and were able to inhibit the 3D growth of both trastuzumab-sensitive and trastuzumab-resistant human breast cancer cells in vitro.

In immunotherapeutic parlance, monoclonal antibodies are classified as passive immunotherapy, whereas vaccines are active immunotherapy [32]. The data of the study illustrate one of the strong points of active immunotherapy, the duration of the antibody response [33]. If one reasons in pharmacological terms, the area under the curve (AUC) of antibodies elicited by ES2B-C001 was several orders of magnitude larger than the AUC that might be obtained by the intermittent administration of exogenous monoclonal antibodies. In perspective, this could also translate into a significant cost effectiveness of vaccinotherapy.

A testament to the functional importance of a long-term antibody response was the prevention of mammary carcinoma onset in HER-2 transgenic mice, in which tumor onset was not observed for more than one year. Mammary carcinogenesis in HER-2 transgenic mice is an unrelenting process that must be continuously kept at bay to avoid cancer onset [31,34,35]. Using a cellular vaccine in a related transgenic model, we found that, even after one year, if the antibody response decreased below a certain level, tumor onset invariably ensued [36]; thus, we plan to go on monitoring anti-HER-2 antibodies in vaccinated long- surviving mice to evaluate the lifetime duration of immune memory.

A persisting anti-tumor immune response would be especially relevant in the therapy of breast cancer, in which tumor (or, more precisely, metastasis) dormancy entails a continuing risk of relapse, even 10 or 20 years after the initial therapy [37,38,39]. In estrogen receptor-positive breast cancer, selective estrogen receptor modulators and aromatase inhibitors are routinely administered for 5 to 10 years, to provide a continuing protection from relapse [40]. The use of vaccines affording long-term protective immune responses might be envisaged in HER-2-positive breast cancer, to prolong the (relatively) short-term efficacy of therapeutic regimes based on monoclonal antibodies. This approach would effectively apply to cancer patients a classical strategy of infectious immunotherapy, in which passive immunotherapy is used to provide protection while the immune response elicited by vaccination is still mounting [41].

A further advantage of ES2B-C001 over monoclonal antibodies was the induction of a broad polyclonal immune response, comprising all IgG isotypes, including IgG2a, IgG2b and IgG3, which were previously found to be instrumental in the protection of transgenic mice from HER-2 expressing tumors [31,42]. In addition to the many anti-tumor functions that could be mediated by the different Fc moieties of the various Ig isotypes, the superior anti-tumor activity of the antibodies elicited by ES2B-C001 could be attributed also to their binding to multiple epitopes of HER-2 p185, resulting in a more complete inactivation of its oncogenic potential [23]. This was demonstrated by their ability to block the 3D growth of the trastuzumab-resistant BT-474 clone C5 in vitro, in the absence of Fc-mediated inhibitory activities.

T cell responses and cytokines elicited by vaccination could also play important roles, not only in anti-tumor immunity, but also in the induction of unwanted side effects that might harm the host, e.g., the cytokine storm syndrome, CSS [43,44]. The analysis of circulating cytokines in immune sera containing high levels of anti-HER-2 antibodies did not reveal a mirroring systemic cytokine response which, coupled with the lack of signs relatable to CSS, indicates that ES2B-C001 did not induce potentially dangerous cytokine responses in mice. Further toxicological studies in other animal species are under way to validate the safety of the vaccine. Evidence of a HER-2-specific T cell response was found in the spleen of vaccinated mice, as evaluated by ELISpot after in vitro restimulation with a mixture of human HER-2 peptides. However, the frequency of IFN-γ producing cell clones remained quite low, around 1/10^4^ splenocytes, even after restimulation. Altogether, the results obtained so far suggest that ES2B-C001 induces strong humoral responses coupled with modest cellular responses. We plan to further monitor the HER-2-specific T cell-induced response and to perform T cell depletion experiments in some of the described in vivo tumor models to better define the role and contribution of T cells subpopulations in the anti-tumor responses studied.

Adjuvants were called by Charles Janeway “the immunologist’s dirty little secret” because vaccines are generally thought to consist of antigens, whereas most vaccines work poorly without adjuvants [45]. In this respect, ES2B-C001 showed a remarkable activity when used alone, for example preventing metastasis outgrowth in the lungs of all FVB mice; however, a few mammary tumors developing in FVB mice and a few metastases in a minor fraction of Delta16 mice were still able to grow, whereas the addition of Montanide boosted the levels of anti-HER-2 total Ig and of all Ig isotypes by one order of magnitude, resulting in an immune response that sterilized tumor and metastasis growth both in FVB and in Delta 16 mice. Therefore, on the basis of these results, Montanide, which is approved for human administration, could be attractive for the formulation of the vaccine for studies in humans [46].

Virus-like particles might at the same time interfere negatively with anti-tumor immunity through the induction of anti-VLP antibodies, an effect that was termed carrier-induced epitopic suppression (CIES) [47]. Results obtained with a previous version of the HER-2 vaccine [24] and with an anti-SARS-CoV-2 vaccine [48] based on the same VLP platform indicate that the AP205-based VLP platform has a net positive effect, strongly enhancing antibody responses in comparison to “naked” antigens. The positive effect is attributable to the very high coupling density of target antigens on the surface of our VLPs [24], leading to robust antigen doses administered repeatedly, which were indicated as key strategies to minimize CIES [47].

Finally, HER-2 positive tumors could develop resistance to HER-2 targeted therapeutic agents, such as antibodies, through the loss of HER-2 expression and the activation of collateral mitogenic pathways [49,50]. Of note, such phenomena occur also in HER-2 transgenic mouse models, which are suitable systems to investigate the onset and evolution of resistance [51]. We did not observe the emergence of resistant tumors in mice repeatedly vaccinated with ES2B-C001+Montanide, not even several months after the last vaccination (see for example Figure 2A and Figure 7B). The results lead us to conclude that the strong and persistent immune responses elicited by the vaccine could eradicate mammary carcinomas driven by HER-2, effectively preventing the onset of resistant variants.

In summary, ES2B-C001 displayed a remarkable preclinical activity in vivo against mouse mammary carcinomas expressing human HER-2, inducing strong specific antibody responses that inhibited human HER-2-positive breast cancer cells in vitro. Thus, ES2B-C001 is a promising candidate vaccine for the therapy of human tumors expressing HER-2. Preclinical results warrant further development towards human clinical studies.

## Figures and Tables

**Figure 1 biomedicines-10-02654-f001:**
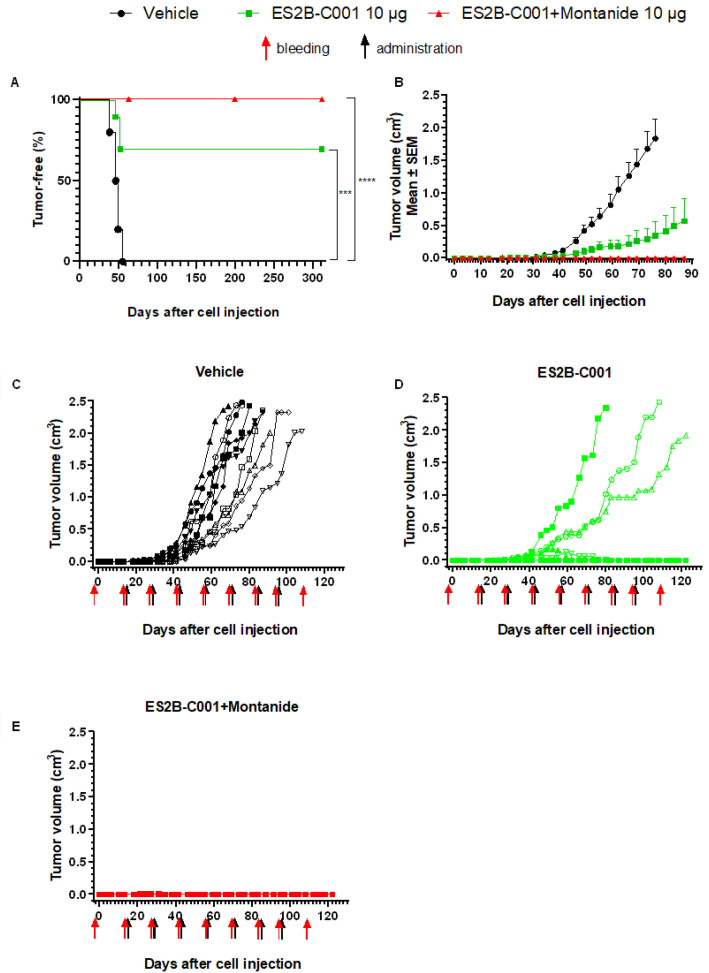
ES2B-C001 therapeutic vaccinations inhibited tumor growth in FVB mice challenged in the mammary fat pad with human HER-2 transgenic mouse mammary carcinoma cells QD. (**A**) Kaplan–Meier tumor-free survival curves. *** *p* < 0.001, **** *p* < 0.0001 by log-rank (Mantel–Cox) test; (**B**) tumor growth curves, each point represents the mean (and SEM) of all mice in each group, the curve interrupts when the first mouse of a group was sacrificed. *p* < 0.05 at least vehicle vs. ES2B-C001 and ES2B-C001+Montanide ISA 51 (referred to as Montanide) from day 49 and *p* < 0.05 ES2B-C001 vs. ES2B-C001+Montanide from day 76 by Tukey’s test; (**C**–**E**) individual tumor growth curves.

**Figure 2 biomedicines-10-02654-f002:**
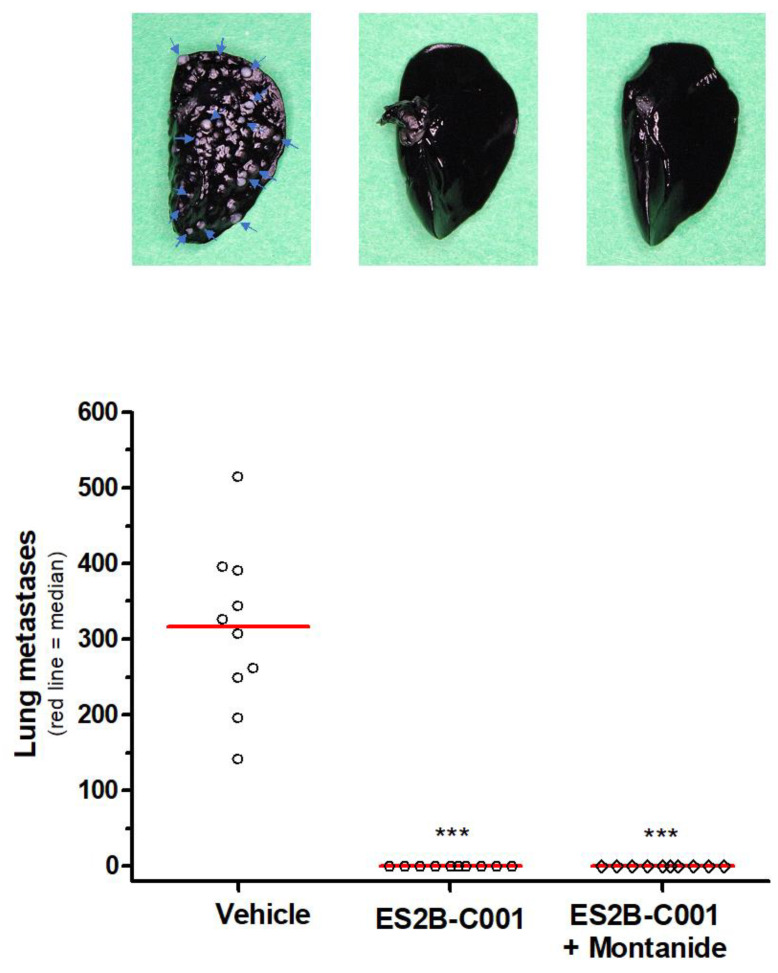
ES2B-C001 therapeutic vaccinations inhibited lung metastases in FVB mice challenged intravenously with QD cells. Lungs were perfused with black India ink to contrast metastatic nodules; pictures show one representative left lung with blue arrows indicating some metastatic nodules (to the left: vehicle; in the middle: ES2B-C001; to the right: ES2B-C001+Montanide). In the graph, each point represents the total number of lung nodules of one mouse, as counted under a dissection microscope; *** *p* < 0.001 vs. vehicle by the Dunn’s non-parametric test.

**Figure 3 biomedicines-10-02654-f003:**
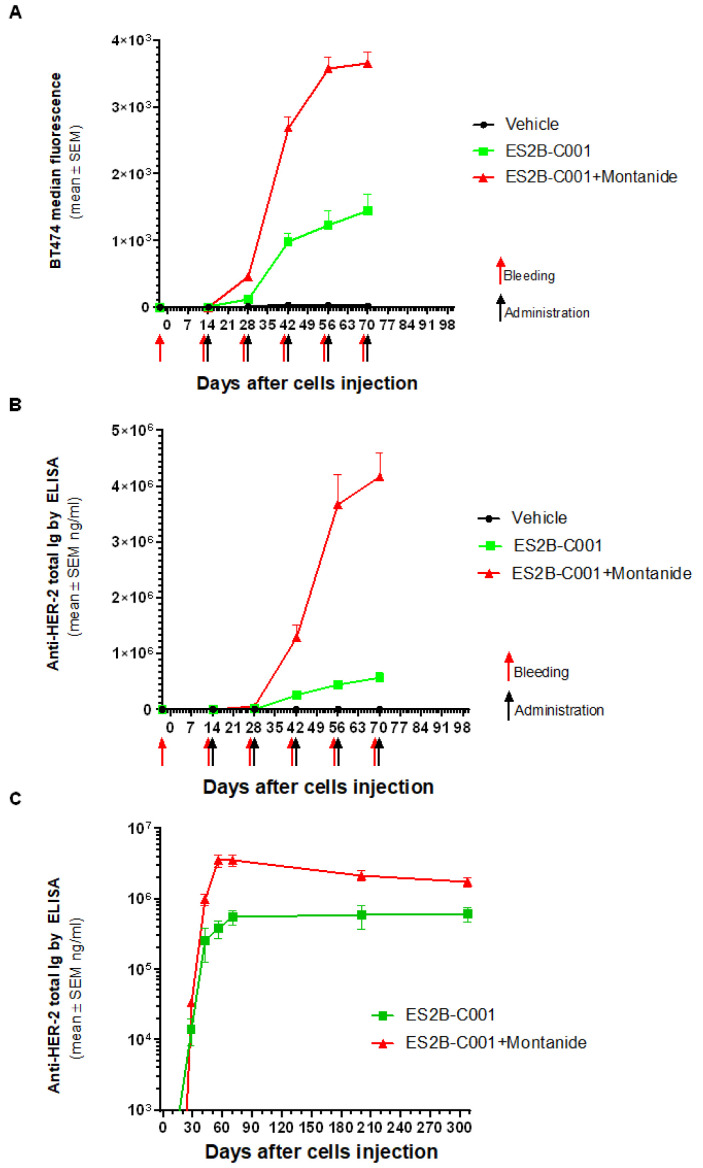
Antibodies elicited in FVB mice by ES2B-C001 therapeutic vaccinations, as measured by flow cytometry of human HER-2^+++^ BT-474 breast cancer cells (**A**) and by ELISA on human HER-2 extracellular domain (**B**). Each point represents the mean (and SEM) of mouse groups shown in Figure 1. ELISA titles were compared by Tukey’s test: *p* < 0.05 vehicle vs. ES2B-C001 at day 70, *p* < 0.0001 vehicle vs. ES2B-C001+Montanide and ES2B-C001 vs. ES2B-C001+Montanid from day 42. (**C**) Long-term persistence of anti-HER-2 antibodies in vaccinated mice after 7 repetitive vaccinations. Each point represents the mean (and SEM) of 3 representative long-surviving mice tested over a period of 10 months.

**Figure 4 biomedicines-10-02654-f004:**
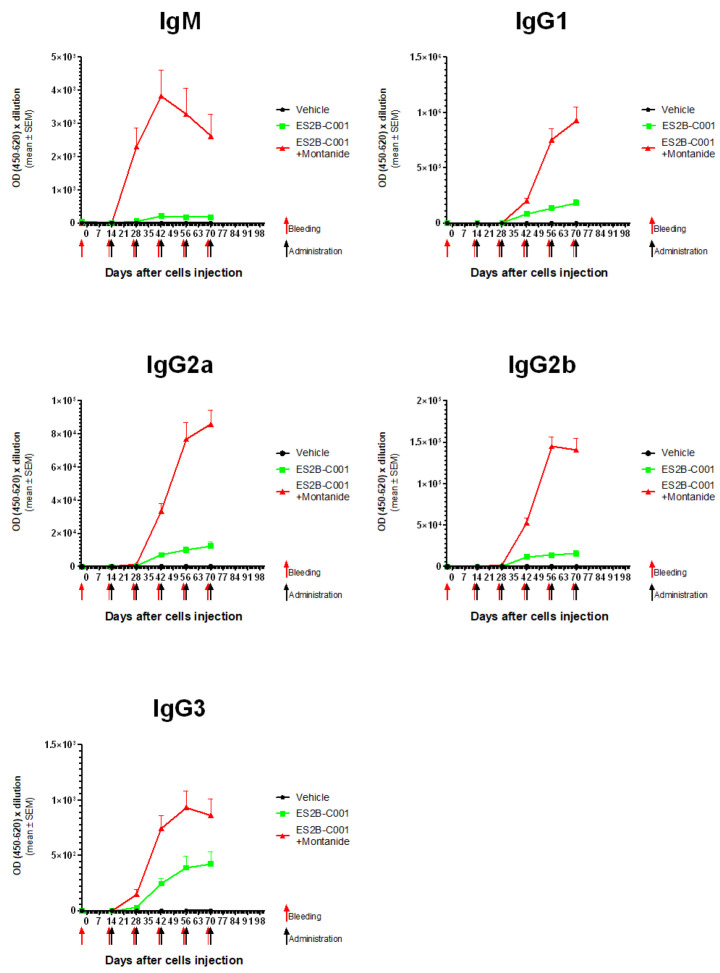
HER-2 specific IgM and IgG isotypes elicited by ES2B-C001 therapeutic vaccinations. Each point represents the mean (and SEM) of mouse groups shown in Figure 1 and Figure 3.

**Figure 5 biomedicines-10-02654-f005:**
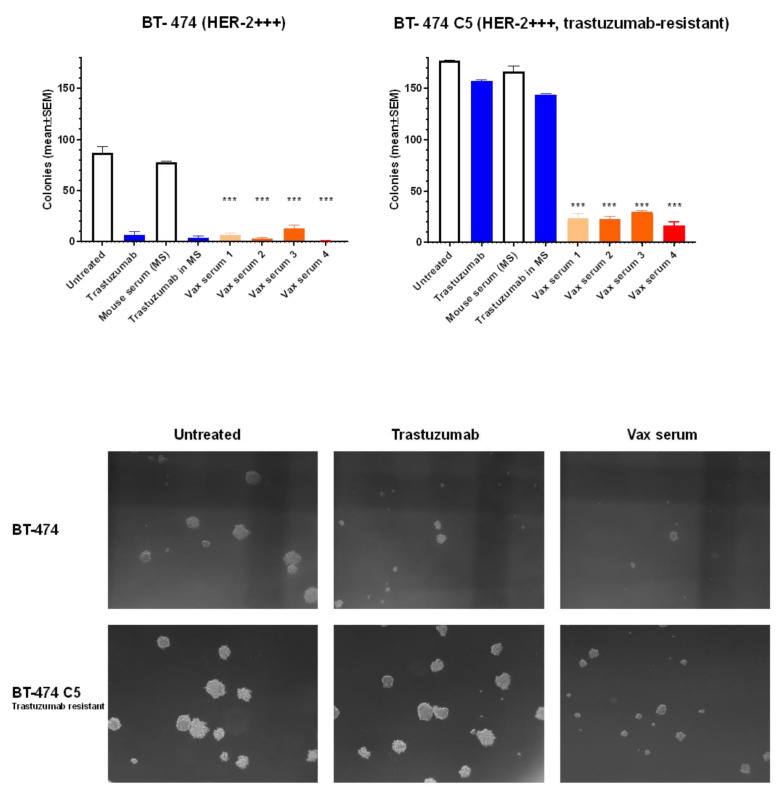
Inhibition of human breast cancer cell 3D agar colony growth by antibodies elicited by ES2B-C001 vaccinations. BT-474 is a HER-2+++ trastuzumab-sensitive cell line, and C5 is a trastuzumab-resistant BT-474 clone. Each bar represents the mean (and SEM) number of colonies larger than 90 µm as counted in two independent cultures with the aid of a micrometer. Vax sera 1–4: sera of 4 previously unchallenged FVB mice vaccinated twice with ES2B-C001 (10–40 µg/mouse i.m. with Montanide) were compared with a pool of naïve mouse sera (MS) as a negative control and benched-marked up against trastuzumab applied in a comparable concentration. *** *p* < 0.001 vs. MS; *p* < 0.05 at least vs. both MS and trastuzumab, Tukey’s test. Representative micrographs of live agar colonies were taken with an inverted microscope (dark-field, 25×).

**Figure 6 biomedicines-10-02654-f006:**
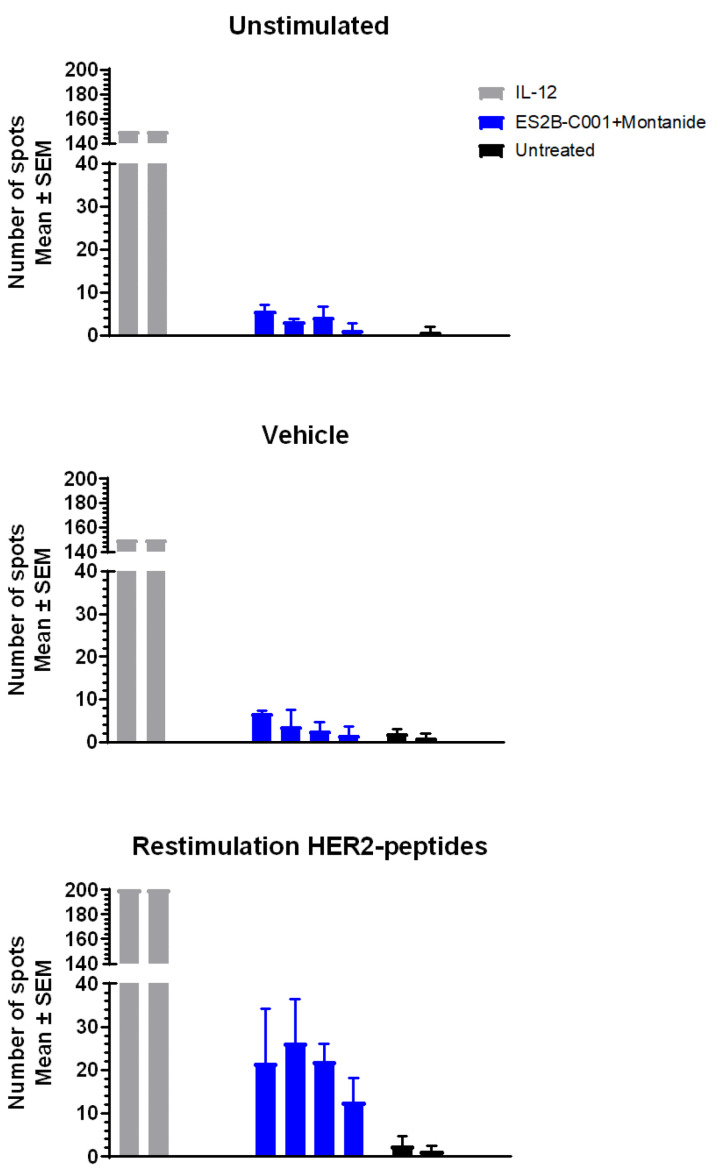
Interferon gamma (IFN-γ) ELISpot analysis of splenocytes from unchallenged FVB mice untreated (*n* = 2, black bars) or previously vaccinated twice with ES2B-C001+Montanide (*n* = 4, blue bars) and restimulated in vitro with a human HER-2 peptide mixture. Positive controls were splenocytes of mice treated three times i.p. with recombinant mouse interleukin 12 (IL-12(p70)) (*n* = 2, gray bars). Each panel represents a different in vitro restimulation. Each bar represents the mean (and SEM) of three wells.

**Figure 7 biomedicines-10-02654-f007:**
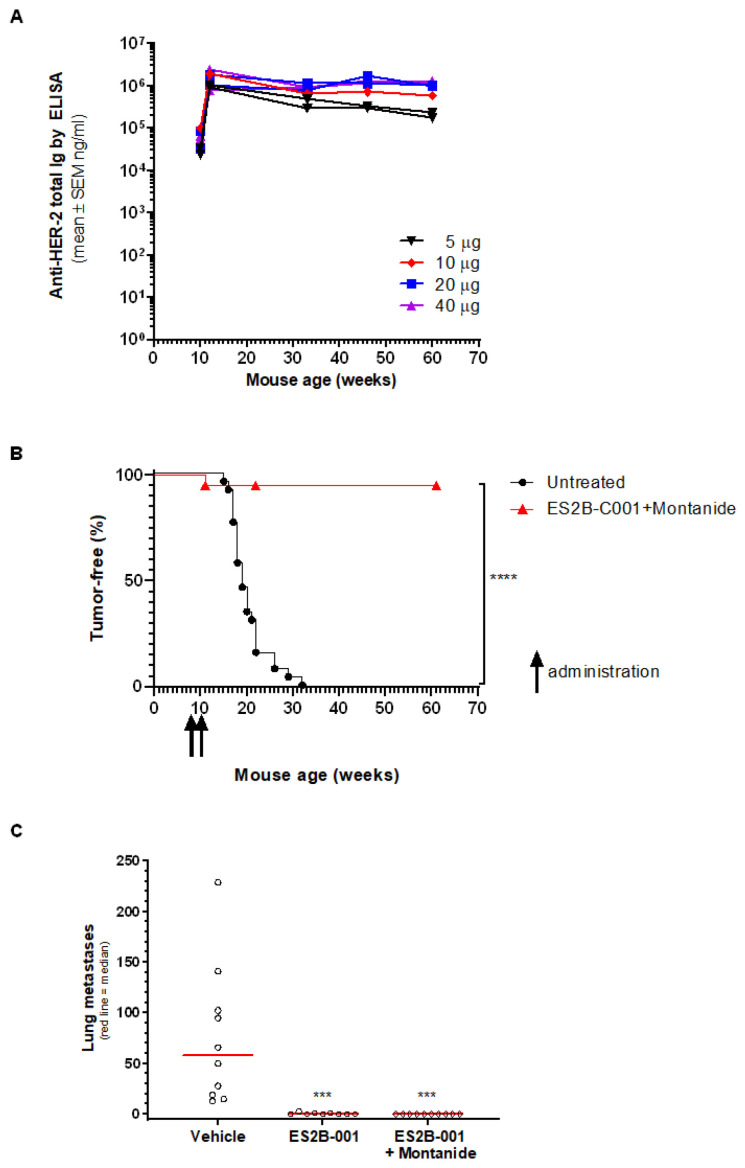
Prophylactic ES2B-C001 vaccination prevented mammary carcinoma onset in HER-2 transgenic Delta16 mice and therapeutic vaccinations inhibited lung metastases outgrowth. (**A**,**B**) A group of 20 mice vaccinated with ES2B-C001+Montanide (5–40 µg/mouse) was followed for more than one year for spontaneous tumor onset and antibody titers. (**A**) Long-term anti-HER-2 antibody response elicited in Delta 16 mice by ES2B-C001 vaccinations, as measured by ELISA; each line represents one mouse. (**B**) The Kaplan–Meier tumor-free survival curve of pooled Delta16 mice vaccinated twice with ES2B-C001 was significantly different (**** *p* < 0.0001 by the log-rank test) from untreated Delta16 mice. (**C**) Therapeutic vaccination of young, tumor-free Delta16 mice challenged i.v. with 0.25 × 10^6^ QD cells; each point represents the total number of lung nodules of one mouse, as counted under a dissection microscope; *** *p* < 0.001 vs. vehicle by the Dunn’s non-parametric test.

## Data Availability

Not applicable.

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
