# Peer review of "Prevention and Therapy of Metastatic HER-2+ Mammary Carcinoma with a Human Candidate HER-2 Virus-like Particle Vaccine"

_biomedicines, 2022, doi:10.3390/biomedicines10102654_

Round 1
Reviewer 1 Report
The manuscript submitted by Ruzzi et al. entitled "Prevention and therapy of metastatic HER-2+ mammary carcinoma with a human candidate HER-2 virus-like particle vaccine" aims to characterize the antitumoral activity of a ES2B-C001, a VLP-based vaccine being developed for human breast cancer therapy, using FVB mice inoculated with HER-2+ mammary carcinoma cells QD. The introduction section is nicely written, although authors should briefly report that the oncogenic role of HER-2 overexpression is also validated in cat, opening a window to additional applications of the tested antibody (e.g. 10.1007/s13277-015-4251-z , 10.18632/oncotarget.7551). The M&M section is very clear allowing the possibility to repeat the experiments. The strongest point of the manuscript is the Results' section. Indeed, all the results are very interesting and relevant for the field. The design of the experiments was very rational, providing a good scientific history. From the results obtained I would like to underline that mice challenged intravenously with QD cells developed a mean of 68 lung nodules in 13 weeks, whereas all mice vaccinated with ES2B-C001+Montanide and 73% of mice vaccinated without adjuvant remained metastasis-free!!!. ES2B-C001 in adjuvant elicited strong anti-HER-2 antibody responses comprising all Ig isotypes. In addition, antibodies inhibited the 3D growth of human HER-2+ trastuzumab-sensitive and -resistant breast cancer cells, with vaccination not inducing a cytokine storm. Thus ES2B-C001 is a very promising candidate vaccine for the therapy of tumors expressing HER-2 and more preclinical tests should be conducted in the near future, in human medicine and feline medicine, also.
In sum, the manuscript is very interesting, providing a nice collection of results towards the characterization of biological role of the ES2B-C001 vaccine using a lab model.
Author Response
Ruzzi et al.
“Prevention and therapy of metastatic HER-2+ mammary carcinoma with a human candidate HER-2 virus-like particle vaccine”
Biomedicines-1919775
Response to Reviewer 1
“The introduction section is nicely written, although authors should briefly report that the oncogenic role of HER-2 overexpression is also validated in cat, opening a window to additional applications of the tested antibody”
Thank you for reviewing our manuscript and your kind comments.
A sentence was added to the introduction to present feline and canine tumors expressing HER-2 (lines 46-48), with relevant references.

Reviewer 2 Report
Dear Authors,
This research showed the candidate human HER-2 vaccine that has been shown a good application prosped for prevention and therapy of mammary carcinoma.
There are two questions about the candidate vaccine
One is whether cancer cells develop resistance if the vaccine is given multiple times during the treatmen.
The other problem is the the candidate human HER-2 vaccine was developed resbased on VLP platform, the body may produce antibodies against VLP when a patient is vaccinated muliple times by HER-2 vaccine. Do the antibodies interfere the vaccine`s effectiveness ?
Author Response
Ruzzi et al.
“Prevention and therapy of metastatic HER-2+ mammary carcinoma with a human candidate HER-2 virus-like particle vaccine”
Biomedicines-1919775
Response to Reviewer 2
Thank you for reviewing our manuscript.
Point 1
“One is whether cancer cells develop resistance if the vaccine is given multiple times during the treatment.”
A paragraph was added to the Discussion on the development of resistance (lines 451-458).
Point 2
“The candidate human HER-2 vaccine was developed resbased on VLP platform, the body may produce antibodies against VLP when a patient is vaccinated multiple times by HER-2 vaccine. Do the antibodies interfere the vaccine`s effectiveness?”
A new paragraph was added to the Discussion to address the interference of anti-VLP antibodies (line 442-450)
